# Enhancing Production of Medium-Chain-Length Polyhydroxyalkanoates from *Pseudomonas* sp. SG4502 by *tac* Enhancer Insertion

**DOI:** 10.3390/polym15102290

**Published:** 2023-05-12

**Authors:** Linxin Song, Ming Wang, Dengbin Yu, Yu Li, Hongwen Yu, Xuerong Han

**Affiliations:** 1International Cooperation Research Center of China for New Germplasm Breeding of Edible Mushrooms, Jilin Agricultural University, Changchun 130118, China; slinxin@yeah.net (L.S.); yudbs@ciac.ac.cn (D.Y.); yuli966@126.com (Y.L.); 2Jilin Province Key Laboratory of Fungal Phenomics, Jilin Agricultural University, Changchun 130118, China; 3School of Life Science and Technology, Changchun University of Science and Technology, Changchun 130022, China; wm0192837465@163.com; 4Tianjin Institute of Industrial Biotechnology, Chinese Academy of Sciences, Tianjin 300308, China; 5Key Laboratory of Wetland Ecology and Environment, Northeast Institute of Geography and Agroecology, Chinese Academy of Sciences, Changchun 130102, China; yuhw@iga.ac.cn

**Keywords:** *Pseudomonas* sp. SG4502, mcl-PHA synthesis, *tac* enhancer, *phaZ* knockout, medium-chain-length polyhydroxyalkanoates

## Abstract

*Pseudomonas* sp. SG4502 screened from biodiesel fuel by-products can synthesize medium-chain-length polyhydroxyalkanoates (mcl-PHAs) using glycerol as a substrate. It contains a typical PHA class II synthase gene cluster. This study revealed two genetic engineering methods for improving the mcl-PHA accumulation capacity of *Pseudomonas* sp. SG4502. One way was to knock out the PHA-depolymerase *phaZ* gene, the other way was to insert a *tac* enhancer into the upstream of the *phaC1*/*phaC2* genes. Yields of mcl-PHAs produced from 1% sodium octanoate by +(*tac-phaC2*) and ∆*phaZ* strains were enhanced by 53.8% and 23.1%, respectively, compared with those produced by the wild-type strain. The increase in mcl-PHA yield from +(*tac-phaC2*) and ∆*phaZ* was due to the transcriptional level of the *phaC2* and *phaZ* genes, as determined by RT-qPCR (the carbon source was sodium octanoate). ^1^H-NMR results showed that the synthesized products contained 3-hydroxyoctanoic acid (3HO), 3-hydroxydecanoic acid (3HD) and 3-hydroxydodecanoic acid (3HDD) units, which is consistent with those synthesized by the wild-type strain. The size-exclusion chromatography by GPC of mcl-PHAs from the (∆*phaZ*), +(*tac-phaC1*) and +(*tac-phaC2*) strains were 2.67, 2.52 and 2.60, respectively, all of which were lower than that of the wild-type strain (4.56). DSC analysis showed that the melting temperature of mcl-PHAs produced by recombinant strains ranged from 60 °C to 65 °C, which was lower than that of the wild-type strain. Finally, TG analysis showed that the decomposition temperature of mcl-PHAs synthesized by the (∆*phaZ*), +(*tac-phaC1*) and +(*tac-phaC2*) strains was 8.4 °C, 14.7 °C and 10.1 °C higher than that of the wild-type strain, respectively.

## 1. Introduction

Polyhydroxyalkanoates (PHAs) have attracted much interest from the scientific community due to their physiochemical properties, including biodegradability and biocompatibility, which are similar to those of petrochemical-based plastics. This intracellular polyester can be synthesized by a variety of bacterial strains under certain deficiency conditions [1] and can serve as a carbon source and energy reserve in bacterial cells [2,3]. PHAs are named according to the number of carbon atoms in their monomeric units: PHAs with C3 to C5 monomer units are commonly called short-chain-length (scl) PHAs, and those with C6 to C14 monomer units are called medium-chain-length (mcl) PHAs [4,5]. Compared with scl-PHAs, mcl-PHAs exhibit elastomer- or adhesive-like properties. Additionally, the scl and mcl copolymerized PHAs have been demonstrated to have better properties and application prospects [6]. PHA synthase (PhaC) is a kind of key enzyme in the PHA synthesis pathway as it determines the monomer structure, composition and molecular weight of PHAs [7]. PhaC can be divided into four categories, namely type I to IV, according to their primary structure, subunit composition and specificity to substrates with different chain length monomers [5,8]. Particularly, type II PhaC is the only polymerase that can synthesize mcl-PHAs from alkane (C6 to C14) precursors. The type II PHA synthases gene locus contains two genes, *phaC1* and *phaC2*, encoding PHA synthase that mainly polymerize the thioesters of coenzyme A (CoA) containing medium- and long-chain PHA monomers to mcl-PHA through β-oxidation or the de novo biosynthesis of fatty acids [9]. Research has shown that PhaC1 and PhaC2 synthases from different *Pseudomonas* sp. strains generate different products through different synthetic pathways. Guo et al. found that engineered bacterium *Pseudomonas mendocina* C7C1 with overexpressing *phaC1* from *P. mendocina* NK-01 could accumulate 9.77% mcl-PHA, while *P. mendocina* C7C2 with overexpressing *phaC2* from the same bacterial strain could produce only 3.67% mcl-PHA. They also found that PhaC1 had higher catalytic activity than that of PhaC2 and was the major PHA synthase for mcl-PHA synthesis in *P. mendocina* [10]. Regarding the *phaC* gene cluster of *Pseudomonas* sp., the *phaZ* gene was found located between the *phaC1* and *phaC2* genes. It has depolymerase activity and can degrade mcl-PHA synthesized by PhaC1 and PhaC2 into aliphatic and aromatic monomers to provide energy for organisms [11,12,13].

To date, studies have shown that numbers of microorganisms can synthesize scl- or mcl-PHAs, but only *Pseudomonas* sp. can accumulate them, due to the presence of two PhaC polymerases, PhaC1 and PhaC2, which have different substrate specificity. Moreover, the incorporation of medium- and long-chain fatty acids into PHA could alter the brittleness and crystallinity of scl-PHAs, making them more elastic. Scl-mcl-PHAs with good elasticity have great application prospects in various industries, such as medicine, packaging and others [14]. However, it is difficult to produce mcl-PHA at a large scale due to the low synthetic efficiency of the strains. The production efficiency of scl-mcl-PHAs could potentially be improved using genetically engineered bacteria generated by deleting or enhancing PHA synthesis-related genes at the molecular level. Two genetic engineering methods have been reported: enhancing the expression of PhaC enzymes and deleting the PHA depolymerase-encoding gene *phaZ* located between *phaC1* and *phaC2*. Akihiko Kondo et al. constructed a recombinant strain by knocking out the depolymerase (*phaZ*) gene and overexpressing genes of poly (3-hydroxyalkanoate) (PHB) biosynthesis to further improve production for PHB [15]. It was reported that the engineered bacteria constructed by overexpressing *phaC1* or *phaC2* genes can improve the synthesis efficiency of mcl-PHA. José G.C. Gomez et al. reported that engineered *Pseudomonas* containing PHA synthases of *Aeromonas* spp. can successfully produce P(3HB-*co*-HA_MCL_) [16]. Nevertheless, recombinant plasmid containing the target gene is unstable and easily lost during the subculture of the host cells. Furthermore, a *tac* enhancer derived from a functional hybrid of trp and lac promoters [17] has been reported to be able to control the expression levels of the target gene [18,19]. For example, Zhang et al. constructed a recombinant strain by integrating a *tac* enhancer on chromosomes to improve L-ornithine production. The recombinant strain had L-arginine production of 25 g/L, which was 63.4% higher than that of the original strain (15.3 g/L) [20]. Fukui et al. demonstrated that the expression of *gfp* under the control of a P*_tac_* promoter was 1.5- to 2-fold higher compared to that of P*_lac_* and P*_phaC_* [21]. The above studies indicate that increasing the expression of the target gene using the *tac* enhancer could improve the yield of the product. Nonetheless, the use of a *tac* enhancer to enhance the expression of PhaC to improve mcl-PHA synthesis in *Pseudomonas* sp. strains has not been verified [22,23,24].

The presence of the *phaZ* gene on the PHA synthesis gene cluster might be one of the reasons for the low efficiency of mcl-PHAs-producing strains containing the Class II PHA synthase genes [25]. The recombinant strain *P. putida* KTMQ01 generated by knocking out the PHA depolymerase *phaZ* gene enhanced the accumulation of mcl-PHA by 20% of cell dry weight (CDW, titer) compared to the mutant *P. putida* KT2442 [26]. However, it has been reported that the yield of mcl-PHA produced by recombinant *P. putida* KT2440 was reduced by 13.9% compared to that produced by the wild-type strain under carbon-limited conditions [27]. The differences in mcl-PHA yield may be derived from different synthetic pathways, PhaC expressions and metabolic balances between different *Pseudomonas* sp. strains, even after the PHA depolymerase *phaZ* gene is deleted.

Thus, this study aims to obtain mcl-PHA with high yields and better thermal stability properties from genetically engineered strains. The strains were generated by inserting a *tac* enhancer on the upstream of *phaC1* or *phaC2* genes to enhance their expression. As an alternative, the *phaZ* gene was deleted from the PHA gene cluster to inhibit PHA degradation in SG4502. Three recombinant strains, *+*(*tac-phaC1*), *+*(*tac-phaC2*) and (∆*phaZ*), were constructed, and their mcl-PHA synthesis ability was compared using the liquid fermentation method. Gene expression was evaluated using real-time quantitative PCR (RT-qPCR) to explain the differences between the expression of mcl-PHA from the *phaC1*, *phaC2* and *phaZ* genes in the three recombinant strains. The composition, structure and molecular weight of the products from the three recombinant strains were determined by hydrogen nuclear magnetic resonance spectroscopy (^1^H-NMR) and gel permeation chromatography (GPC). Moreover, the thermal properties of the products were determined by differential scanning calorimetry (DSC) and thermogravimetric analysis (TGA).

## 2. Materials and Methods

### 2.1. Strains and Plasmids

*Pseudomonas* sp. SG4502 (NCBI: txid547985) was provided by a laboratory at Hokkaido University, Japan. *Escherichia coli* DH5α and *E. coli* JM109 were preserved in the laboratory. Plasmids pUC19 and pCDFDuet-1 were purchased from MiaoLing Company (Wuhan City, China). Plasmid pK18 (containing the strong *tac* enhancer) was provided by Professor Guangming Xiong, Kiel University, Germany. All strains and plasmids used in this study are listed in Appendix A.

### 2.2. Reagents and Media

Tryptone, yeast extract, NaCl, glycerol, Nile red, dimethyl sulfoxide (DMSO), agarose, absolute ethanol, agar powder, NaOH, Na_2_EDTA·2H_2_O, Tris and glacial acetic acid were purchased from Beijing Chemical Plant (Beijing, China). Ampicillin, Kanamycin, Streptomycin, chloramphenicol and tetracycline were purchased from Baotaike (Shijiazhuang, China). X-gal and IPTG were purchased from Shanghai QianChen Biotechnology Co., Ltd. (Beijing, China). All chemical reagents were of analytical grade. The concentration of antibiotics used in this study was as follows: C_ampicillin_ = 100 mg·mL^−1^, C_kanamycin_ = 50 mg·mL^−1^, C_streptomycin_ = 50 mg·mL^−1^, C_tetracycline_ = 50 mg·mL^−1^ and C_chloramphenicol_ = 25 mg·mL^−1^.

A Lysogeny Broth (LB) medium was prepared by vigorously mixing 1.0 g of sodium chloride, 1.0 g of tryptone and 0.5 g of yeast extract in 100 mL of ultrapure water using a magnetic stirrer [28]. The medium was used to cultivate wild-type SG4502 cells (45 °C, 180 rpm).

In the pre-culture of all recombinant bacteria (45 °C, 180 rpm), a nutrient broth (NB) was used. The broth was prepared by thoroughly mixing 24.0 g of NB broth and 3 g of yeast extract in 1 L of ultrapure water. Additionally, 1.5 g of agar powder was added to solidify the NB medium.

An inorganic salt medium (MS) was prepared by dissolving 3.6 g of Na_2_HPO_4_, 1.5 g of KH_2_PO_4_, 1.0 g of NH_4_Cl, 0.2 g of MgSO_4_·7H_2_O and 1 mL of trace elements in 1 L of ultrapure water. The bottle was sealed with a sealing film after the solution was clarified [29]. After a carbon source was added, PHA for fermentation and cultivation (45 °C, 180 rpm).

All the above media were divided into tubes and sterilized by autoclaving at 121 °C and 1 atm for 20 min.

### 2.3. Experimental Methods

#### 2.3.1. Construction of Recombinant Strain ∆*phaZ*

The PHA depolymerase-encoding gene *phaZ* located between the *phaC1* and *phaC2* genes was knocked out from the genome of wild-type SG4502 by inserting the *smr* gene into the *phaZ* gene to interrupt its transcription. The construction process of ∆*phaZ* is shown in Figure 1 and has been described previously [25]. Using the genomic DNA of wild-type SG4502 as a template, PCR amplification of the *pC1ZC2* (C1-Z-C2) fragment was carried out using HS DNA Polymerase (TOYOBO, Osaka, Japan), an upstream primer, pC1ZC2-F, and a downstream primer, pC1ZC2-R. The amplified fragment contained about 500 bp of *phaC1*, the full-length sequence of *phaZ* and about 500 bp of *phaC2*; the total sequence length was 2185 bp (C1-Z-C2). *pC1ZC2* (C1-Z-C2) was digested by the *Hin*dIII and *Eco*RI restriction enzymes and then inserted into pUC19 plasmid treated with the same restriction enzymes using EasyGeno Rapid Recombinant Cloning Kit (TIANGEN, Beijing, China). Then, the recombinant plasmid pUC19-*C1ZC2* was transferred into *E. coli* JM109, and colonies containing the recombinant plasmid pUC19-*C1ZC2* were screened using blue-white screening. An *smr* gene fragment cloned from pCDFDuet-1 using Smr-F and Smr-R primers was inserted into the *phaZ* sequence of the recombinant plasmid pUC19-*C1ZC2* using the *Sac*II restriction enzyme to prevent the translation of the *phaZ* gene. Afterwards, the reaction solution was transferred into *E. coli* JM109 using the heat shock method, and bacterial colonies containing recombinant plasmid pUC19-*C1ZC2*-*smr* were screened using Ampicillin and Spectinomycin resistance screening. Moreover, the recombinant plasmid was sequenced by Jilin KuMei company (Changchun, China); after that, it was transferred into competent cells of wild-type SG4502 by electroporation to knockout the *phaZ* gene through homologous recombination. *Hin*dIII and *Eco*RI dual digestion reactions were used to identify the deletion mutant. The recombinant strain ∆*phaZ* was obtained after the transcriptional expression of *phaZ* was inhibited by *smr* insertion. The results were validated by PCR identification.

#### 2.3.2. Construction of *phaC1* and *phaC2* Gene-Overexpression Recombinant Strains

In this study, pK18 plasmid containing the strong *tac* enhancer was used to overexpress PHA polymerases PhaC1 and PhaC2, from SG4502. The construction process of the recombinant plasmids is shown in Figure 2a. The *phaC1* and *phaC2* genes were amplified by PCR from the genomic DNA of wild-type SG4502 extracted using a Bacterial Genomic DNA Extraction Kit (TIANGEN, Beijing, China). Upstream and downstream primers (listed in Appendix A) of *phaC1* (phaC1-F phaC1-R) and *phaC2* (phaC2-F phaC2-R)*,* respectively, were used for PCR amplification. Subsequently, the amplified fragments of the *phaC1* and *phaC2* genes were inserted into the vector pK18-*tac* at *Kpn*I and *Bsp*119I enzyme digestion sites, repectively, by the NEBuilder HiFi DNA assembly method using an EasyGeno Rapid Recombinant Cloning Kit (TIANGEN, Beijing, China). The relative position of *phaC1/phaC2* and *tac* in pK18-*tac*-*phaC1* and pK18-*tac*-*phaC2* are shown in Figure 2b. The position of the *tac* promoter is presented in Figure 2c. In the wild-type, the *phaC1* and *phaZ* genes were closely connected, and then there was a spacer sequence between *phaZ* and *phaC2.* The *tac* promoter was in the upstream of *phaC1* and *phaC2* in +(*tac-phaC1*) and +(*tac-phaC2*), respectively. The recombinant plasmids were transferred into *E. coli* DH5α using the heat shock method for plasmid replication, and the plasmid was then extracted from the *E. coli* DH5α cells. Desalted and dried recombinant plasmid (5 μL) extracted from *E. coli* DH5α in the previous step was added into 100 μL of competent wild-type SG4502 cells. The mixture was transferred into an electric shock cup, in which it was treated with a voltage of 1900 V and a shock time of 5 ms. Moreover, the shock resistance was 200 Ω after the tube was placed on ice for 20 min [30]. The bacterial liquid was collected by centrifugation and then resuspended and smeared on a plate containing medium with different antibiotics (Kan, 50 mg·mL^−1^), and the plate was incubated at 37 °C until colonies were observed. After that, the recombinant strains containing *tac-phaC1* and *tac-phaC2* were screened by PCR using q-phaC1-F, q-phaC1-R, q-phaC2-F and q-phaC2-R primers. Finally, the recombinant plasmids were sequenced by Jilin KuMei company (Changchun, China).

#### 2.3.3. Fermentation of Recombinant Strains and Extraction of Products

Wild-type and recombinant strains were inoculated in 100 mL of NR medium and cultured overnight to prepare a seed culture. The seed culture was inoculated into 100 mL of MS medium containing 1% sodium octanoate and cultured at 45 °C and 180 rpm for 72 h. Three parallel experiments were conducted for each strain. The bacterial cells were collected by centrifugation at 4 °C at 5000 rpm for 10 min and then washed thrice with distilled water. The CDW of the bacteria was weighed after being freeze-dried (FD-1B-80). Then, PHA was extracted using the chloroform-methanol precipitation method, as has been described elsewhere [31].

#### 2.3.4. Nile Red Fluorescent Staining of Recombinant Strains

Nile red is a lipophilic fluorescent dye that quickly, sensitively and reliably detects lipid components in cells; it emits a fluorescent signal after binding to lipid substances such as wax esters, triacylglycerols and various fatty acids [32]. To macroscopically investigate the accumulation of PHA in different strains, the cellular contents of PHA in the culture described in Section 2.3.3 were tracked at different time points (6, 12, 24, 36, 48 and 72 h) using Nile red fluorescence staining. At each time point, 1 mL of bacterial liquid was withdrawn, centrifuged and then washed twice with distilled water. Next, the bacterial cells were resuspended in ultrapure water (1.5 mL) and then mixed with 3 μL of Nile red solution. The mixture was incubated at 37 °C in the dark for 15 min, and its fluorescence emission at 530 nm was determined using a Biotek’s fluorescence assay microplate reader to initially determine the PHA synthesis ability of wild-type SG4502, ∆*phaZ* mutant, recombinant +(*tac-phaC1*) and +(*tac-phaC2*) strains.

#### 2.3.5. RT-qPCR of *phaC1*, *phaC2* and *phaZ* Genes in Different Strains

The strains were cultured in 50 mL of NB medium at 45 °C and 180 rpm for 72 h. Then, the bacteria were collected by centrifugation at 12,000 rpm at 4 °C for 2 min and then diluted to 1 × 10^9^ CFU/mL. The total RNA of the four strains was extracted using a cultured cell/bacterial total RNA extraction kit (TIANGEN, Beijing, China) and used as a template in the reverse transcription reaction to synthesize cDNA. The cDNA was used in the real-time measurement of the transcription levels of the *phaC1*, *phaC2* and *phaZ* genes using q-phaC1-F/R, q-phaC2-F/R and q-phaZ-F/R primer pairs, respectively (Appendix A).

RT-qPCR was performed using Super Real Fluorescence Quantity Premix Reagent Color Edition (TIANGEN, Beijing, China) on a Mastercycler ep realplex real-time PCR system (Eppendorf, Hamburg, Germany). RT-qPCR conditions were as follows: pre-denaturation at 95 °C for 15 min, followed by denaturation at 95 °C for 10 s and extension at 63.8 °C for 32 s for a total of 40 cycles. Each assay was performed in triplicate. Moreover, the relative gene expression level of the recombinant strains was normalized to that of the wild-type strain, and the calculations were carried out using the 2^−ΔΔCt^ method against the 16S rRNA gene, which was an internal reference.

#### 2.3.6. Compositional and Structural Analysis of Accumulated Products

About 5 to 10 mg of the extracted product was dissolved in 0.5 mL of deuterated chloroform. ^1^H-NMR spectroscopy of polymer PHA was conducted on a Bruker MSL 400 spectrometer (400 MHz) at 90 °C, a pulse of 4 ms, a spectral width of 3000 Hz and a repetition rate of 4 s.

20 mg of purified PHA product was added into 10 mL of deuterated chloroform and let stand at room temperature for more than 12 h until thoroughly mixed. After being filtered through a 0.22 μm membrane, the sample was subjected to GPC detection using a Shimadzu gel chromatograph at 30 °C. HPLC grade chloroform was used as the mobile phase, and polystyrene was used as the standard sample. The flow rate was 1 mL·min^−1^, and the injection volume was 50 μL.

#### 2.3.7. Thermal Property Analysis

The melting point and the glass transition temperature of the PHA product were detected on a differential scanning calorimeter using a sample with a mass of about 20 mg. Nitrogen was used as a protective carrier gas, of which the flow rate was set at 50 mL·min^−1^. The sample was reheated from −50 °C to 220 °C at a heating rate of 10 °C·min^−1^.

Thermogravimetric data of the PHA product were determined on a synchronous thermal analyzer STA449C (NETZSCH, Bayern, Germany) using about 20 mg of sample. Nitrogen was used as a protective carrier gas; its flow rate was set at 10 mL·min^−1^. The temperature was increased from 50 °C to 350 °C at a rate of 10 °C·min^−1^.

## 3. Results and Discussion

### 3.1. Construction and Identification of the Knockout Strain ∆phaZ

Firstly, to construct the recombinant strain (∆*phaZ*), the *C1ZC2* gene was amplified by PCR. The results from electrophoresis (Appendix A) showed that there was a clear band at around 2000 bp, which corresponds to the theoretical size (2185 bp). The amplified *C1ZC2* gene fragment was digested by the *Hin*dIII and *Eco*RI sites and then inserted into the pUC19 plasmid linearized by the same restriction enzymes, from which the recombinant plasmid pUC19-*C1ZC2* was generated, as can be seen in Appendix A. In correspondence with the expected size, the size of the recombinant plasmid was about 4700 bp, indicating that the recombination was successful.

Furthermore, the *smr* gene was inserted into the *phaZ* gene to prevent its transcription. The electrophoresis results (Appendix A) of the *smr* gene cloned from pCDFDuet-1 plasmid showed a band at about 1025 bp, which corresponds to the theoretical size (1000 bp). The cloned *smr* gene was inserted into the pUC19-*C1ZC2* plasmid to construct a pUC19-*C1ZC2*-*smr* recombinant plasmid using the *Sac*II enzyme. The electrophoresis (Appendix A) of enzyme digestion products of pUC19-*C1ZC2*-*smr* plasmid showed two obvious bands: one was at about 3500 bp, and the other was near 1900 bp. These bands were in line with the theoretical sizes of the plasmid cleaved by the enzyme. Moreover, the sequencing results demonstrated that the similarity between the recombinant plasmid sequence and the ideal sequence was 100% (Appendix A).

The constructed pUC19-*C1ZC2*-*smr* was transferred into competent wild-type SG4502 to generate the *phaZ-*knockout recombinant strain and screened based on Amp and Smr double resistance. PCR results showed that the *smr* gene could be amplified from the ∆*phaZ* but not from the wild-type strain (Figure 3). Thus, these results showed that the deletion mutant ∆*phaZ* was successfully constructed.

### 3.2. Construction and Identification of Recombinant Strains +(tac-phaC1) and +(tac-phaC2)

PCR amplification results of *phaC1* and *phaC2* showed two obvious bands at approximately 1700 bp, both of which were consistent with the theoretical sizes of both *phaC1* and *phaC2* (Appendix A). The recombinant plasmid containing the target gene *phaC1*/*phaC2* was obtained by ligating with pK-18 vector. After *phaC1*/*phaC2* was verified by colony growth, PCR amplification (Appendix A) and sequence similarity (Appendix A), the recombinant plasmid pK18-*phaC1*/*phaC2* was electroporated into competent wild-type SG4502 cells, and the recombinant strains were screened through Kanamycin resistance. The electrophoresis results indicated that the recombinant strains containing the *tac* enhancer +(*tac-phaC1*)/+(*tac-phaC2*) had been successfully constructed (Appendix A).

### 3.3. Product Accumulation in Recombinant Strains

In order to evaluate the PHA accumulation capacity of the three recombinant strains, cells were stained with Nile red at different time points during cultivation (Figure 4a) [33]. As time went on, the magnitude of fluorescence signal could be ranked as follows: +(*tac-phaC2*) > ∆*phaZ* > wild-type SG4502 > +(*tac-phaC1*). The results above indicate that under the same conditions, the recombinant strains +(*tac-phaC2*) and ∆*phaZ* have a better ability to synthesize PHA than wild-type SG4502. Similarly, both the CDW and the product yield of the three strains were compared (Figure 4b). The CDW of +(*tac-phaC1*), +(*tac-phaC2*), ∆*phaZ* and wild-type SG4502 was 0.57, 1.83, 1.44 and 1.18 g/L, respectively. This result showed that the growth rate of all genetically engineered bacteria was higher than that of the wild-type, and the genetic engineering had no effect on the growth of the strains. Additionally, among all strains, the +(*tac-phaC2*) strain had the highest growth rate. The product yield of +(*tac-phaC1*), +(*tac-phaC2*), ∆*phaZ,* and wild-type SG4502 was 0.013, 0.04, 0.032 and 0.026 g/g, respectively. The products yield was 50.0% lower in *+*(*tac-phaC1*) and 53.8% and 23.1% higher in *+*(*tac-phaC2*) and ∆*phaZ* than the wild-type. The product yield was consistent with the results of CDW and Nile red fluorescence staining. When comparing with the wild type, *+*(*tac-phaC2*) showed a better proliferation ability and PHA synthesis ability than *+*(*tac-phaC1*). This was probably because the *tac* position in the PHA synthase gene cluster had an influence on gene transcription and protein expression. Additionally, the ∆*phaZ* strain had a higher production yield than the wild type, which could avoid the degradation of PHA synthesized during the whole culture period. To explain the reasons for the different product yields from different recombinant strains, the transcriptional levels of *phaC1*, *phaZ* and *phaC2* genes in three strains were investigated.

### 3.4. Transcription Level of phaC1, phaZ, and phaC2 Genes in Recombinant Trains

To explore why the product yields were different, the transcription levels of the *phaC1*, *phaC2* and *phaZ* genes in the three recombinant strains were investigated by RT-qPCR (Figure 5). Compared with that of wild-type SG4502, the relative transcription level of the *phaC1* and *phaZ* genes in +(*tac*-*phaC1*) was higher, while that of the *phaC2* gene was slightly lower. For +(*tac-phaC2*), the relative transcription level of *phaC1* and *phaC2* was considerably higher, while that of *phaZ* was lower, compared to that of wild-type SG4502. These results suggest that the insertion position of *tac* can directly affect the transcription level of genes related to PHA synthesis. When the *tac* enhancer was inserted at the upstream of *phaC1,* the transcription level of *phaZ* (located downstream of *phaC1* in the same gene cluster) and *phaC1* significantly increased. At the same time, the transcription level of *phaC2* located at the downstream of *phaZ* in the same gene cluster was not more impacted due to its long distance from *tac*. The transcription level of a gene directly affects its protein expression level. Considering the transcription levels of PHA synthase genes in +(*tac-phaC1*), the expression levels of PhaC1 and PhaZ proteins should be significantly increased, while that of PhaC2 protein should not drastically change; this can disrupt the balance between the synthesis and degradation of products, causing the product yield to decrease. Our results were consistent with those from a previous study in which the overexpression of *phaC* caused the expression of *phaZ* to be higher and the PHA yield to be lower compared with those of the wild type [34,35]. On the other hand, in +(*tac*-*phaC2*), the *tac* enhancer located upstream of *phaC2* at a distance from *phaZ* (which was located between *phaC1* and *phaC2*) enhanced the transcription level of *phaC1* and *phaC2* but decreased the transcription level of *phaZ*. Higher transcription levels of *phaC1* and *phaC2* and a lower transcription level of *phaZ* can directly affect the expression level of the corresponding proteins, resulting in an increase in the product yield in +(*tac*-*phaC2*) compared to that in the wild-type. In brief, the *tac* enhancer can influence the transcription level of *phaC* genes and improve the yield of mcl-PHA in genetically engineered bacteria, and the location and distance of *tac* from the target genes in the same gene cluster play important roles in the change.

Finally, regarding knockout strains ∆*phaZ*, the transcription level of the *phaC1* gene was not affected, while *phaC2* was significantly increased and *phaZ* was almost not transcribed. During the culture process of ∆*phaZ*, the low expressed PhaZ should greatly reduce the decomposition ability of synthesized mcl-PHA, coupled with the high expression of PhaC2, resulting in a significant increase in the product yield. These results showed that a high transcription level of *phaC2* is crucial for mcl-PHA synthesis in wild-type SG4502. Furthermore, in SG4502, PhaC1 was active for short chain fatty acids such as 3HB, 2HB and synthesized scl-PHA [13]. For wild-type SG4502, PHA with different properties could be an effective product by the tendentious change in the transcription or expression level of *phaC1/phaC2* based on different substrate specificity using the molecular method.

### 3.5. Structure, Composition, and Molecular Weight of mcl-PHA

The structure, composition and molecular weight of products synthesized from wild-type SG4502, +(*tac*-*phaC1*), +(*tac*-*phaC2*) and ∆*phaZ* were analyzed by ^1^H-NMR and GPC.

#### 3.5.1. ^1^H-NMR

The ^1^H-NMR results for the four samples are shown in Figure 6. The peaks centered at 0.86, 1.28, 1.50, 2.50 and 5.11 ppm corresponded to CH_3_ (a1, a2, a3), (CH_2_)_n_ (b1, b2, b3), CH_2_ (branched chain) (c1, c2, c3), CH_2_ (main chain) (e1, e2, e3) and CH (d1, d2, d3) of the three components (3HO, 3HD and 3HDD) in the polymers, respectively [11]. This result revealed that the transcription levels of *phaC*1, *phaC2* and *phaZ* had no impact on the component of monomer units in mcl-PHA synthesized.

The ^1^H-NMR results indicated that the polymer synthesized by recombinant bacteria using sodium octanoate as the sole carbon source contained the same monomers as wild-type SG4502. Compared to the wild type, the production of copolymers synthesized by +(*tac-phaC2*) increased by 53.8%, while +(*tac-phaC1*) decreased by 50.0%. The reason for this is attributed to the substrate specificity of the *phaC1* and *phaC2* genes. *phaC1* has activity on short chain fatty acids, while *phaC2* has activity on medium and long chain fatty acids [36]. Using sodium octanoate as the sole carbon source, enhanced transcription of *phaC2* leads to an increase in protein expression, which in turn leads to an increase in polymer production synthesized by +(*tac-phaC2*). However, +(*tac-phaC1*) with higher transcription of *phaC*1 and *phaZ* showed a sharp decrease in the yield of the synthesized copolymer because of the low substrate specificity of PhaC for sodium octanoate. In addition, ∆*phaZ* are similar to +(*tac phaC2*); the deletion of *phaZ* gene leads to a significant increase in *phaC2* transcription, resulting in a significant increase in polymer production [13].

#### 3.5.2. GPC

The molecular weight distribution of the synthesized mcl-PHA was determined by GPC (Table 1). The M_n_ of mcl-PHAs from the +(*tac-phaC1*), +(*tac-phaC2*) and ∆*phaZ* strains was higher than that of the wild-type strain. Theoretically, the PDI of molecular weight indicates the homogeneity of the polymer and is the factor directly affecting its properties. In this study, the PDI of mcl-PHAs from recombinant strains was one-half lower than that of mcl-PHAs from the wild type, indicating that mcl-PHAs from the former were more evenly distributed than those from the latter, which is consistent with the molecular weight PDI of similar polymers reported previously [4,10]. Thus, we may conclude that the change in PDI of mcl-PHAs from three recombinant strains might have a positive impact on their thermal properties.

### 3.6. Thermal Property of mcl-PHA

#### 3.6.1. DSC

To investigate the influence of PDI on the thermal properties of mcl-PHAs, DSC and TGA were carried out. The melting temperature (*T_m_*) and glass transition temperature (*T_g_*) of mcl-PHAs from +(*tac-phaC1*), +(*tac-phaC2*), ∆*phaZ* and wild-type SG4502 were 62.73 °C and −34.08 °C (Figure 7a), 64.83 °C and −40.90 °C (Figure 7b), 63.94 °C and −36.69 °C (Figure 7c) and 64.67 °C and −38.38 °C (Figure 7d), respectively. These results indicated that the *T_g_* and *T_m_* of mcl-PHA synthesized by the recombinant strains were similar to those of mcl-PHA synthesized by the wild-type strain, which is consistent with the findings reported in the literature [37].

#### 3.6.2. TGA

As can be seen in Figure 8, the maximum decomposition temperature of PHA from wild-type SG4502, +(*tac-phaC1*), +(*tac-phaC2*) and ∆*phaZ* was 356.7 °C, 365.1 °C, 371.4 °C and 366.8℃, respectively. Additionally, the decomposition temperature of mcl-PHA synthesized by +(*tac-phaC1*), +(*tac-phaC2*) and ∆*phaZ* was 8.4 °C, 14.7 °C and 10.1 °C, respectively, higher than that of the wild-type strain. Moreover, the decomposition temperature of mcl-PHA from the recombinant strains increased throughout the assay. The increase in thermal stability may be due to the irregular chain length of the copolymer, which could result in a high degree of crystallization of side chains; consequently, the thermal stability of the polymer improved [38]. Combined with the results on molecular weights, the PDI of mcl-PHA from +(*tac-phaC2*) was lower compared with that of mcl-PHA from wild-type SG4502 and +(*tac-phaC1*). Thus, it can be inferred that the polymer synthesized by the recombinant strains is more evenly distributed and more stable than that synthesized by the wild-type strain.

## 4. Conclusions

In sum, the recombinant strains of SG4502, +(*tac-phaC2*) and ∆*phaZ* could efficiently improve the yield of mcl-PHA. Compared with the *phaZ* gene deletion method, the *tac* insertion method is more efficient in synthesizing products. mcl-PHAs synthesized by *+*(*tac-phaC1*)/+(*tac-phaC2*) and ∆*phaZ* contain 3HO, 3HD and 3HDD monomer units that are comparable to those from wild-type SG4502. In addition, the PDI of mcl-PHAs from the three recombinant strains was lower than that of mcl-PHAs from the wild-type strain. The *T_m_* and *T_g_* of mcl-PHAs synthesized by all the recombinant strains were within the known *T_m_* and *T_g_* ranges of mcl-PHAs. The decomposition temperature of mcl-PHAs from the recombinant strains was higher than that of mcl-PHAs from the wild-type strain, indicating that they have higher thermal stability. Finally, the *tac* enhancer insertion method is a simpler and more flexible method for regulating the transcription and expression of the target gene compared to the gene knockout method. Together, this study presents a strategy for producing genetically engineered bacteria that can highly efficiently produce mcl-PHA.

## Figures and Tables

**Figure 1 polymers-15-02290-f001:**
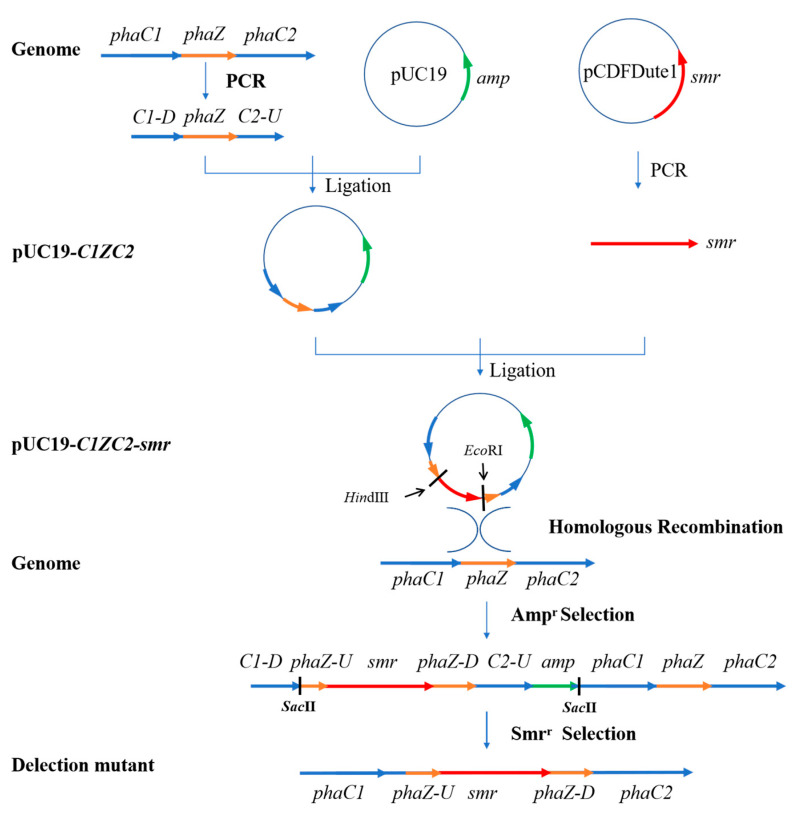
Construction process of knockout mutant ∆*phaZ*.

**Figure 2 polymers-15-02290-f002:**
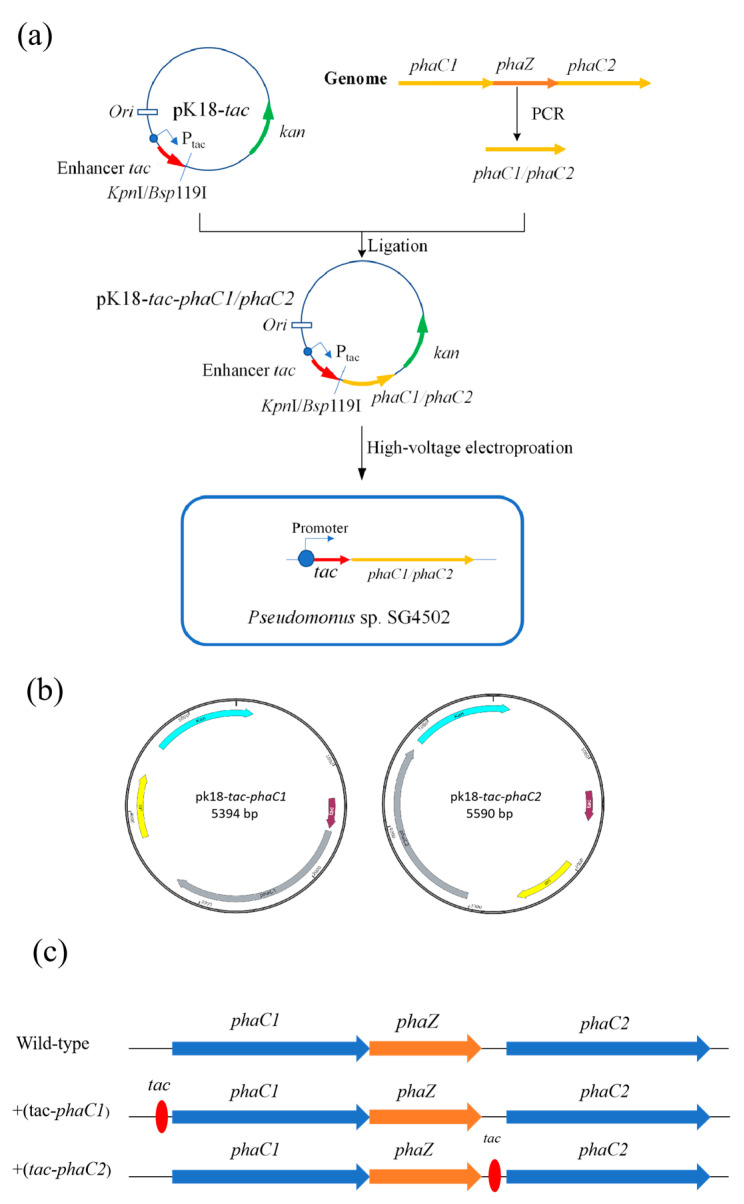
Overexpression plasmid construction. (**a**) +(*tac-phaC1*)/+(*tac-phaC2*) plasmid construction process. (**b**) Maps of recombinant plasmids. (**c**) the position of *tac* enhancer in the +(*tac-phaC1*)/+(*tac-phaC2*).

**Figure 3 polymers-15-02290-f003:**
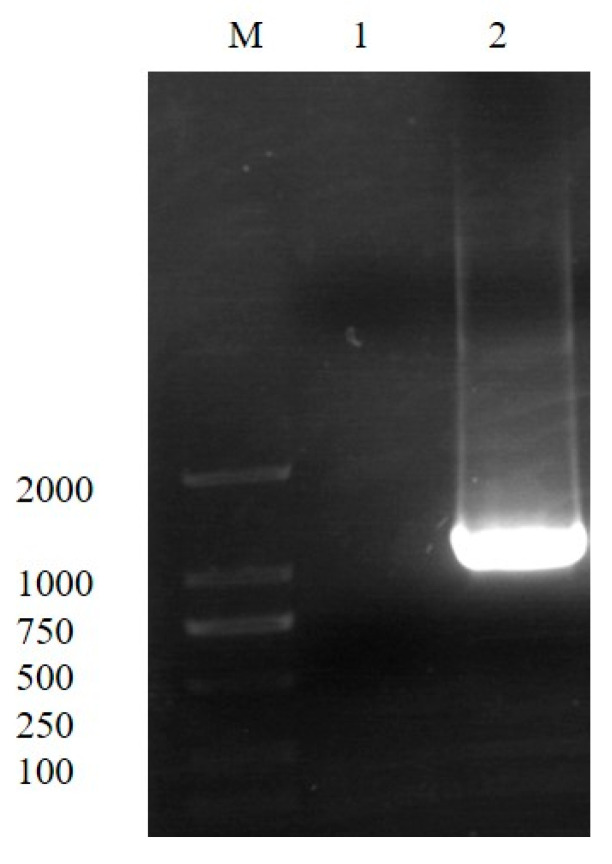
PCR amplification of *smr* in wild-type SG4502 (1) and the knockout strain ∆*phaZ* (2).

**Figure 4 polymers-15-02290-f004:**
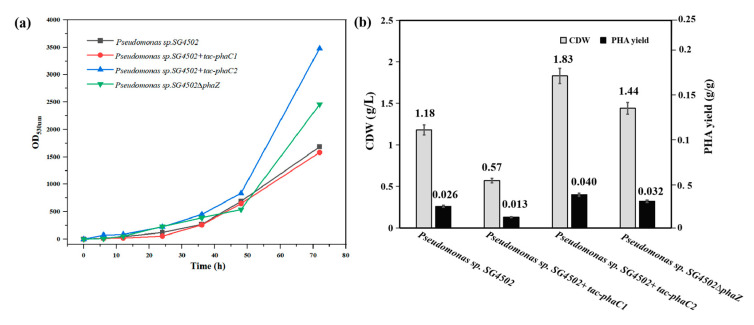
(**a**) Nile red fluorescent staining experiment (Results of *Pseudomonas* sp. SG4502, *Pseudomonas* sp. SG4502 *+ tac-phaC1*, *Pseudomonas* sp. SG4502 + *tac-phaC2* and *Pseudomonas* sp. SG4502∆*phaZ* were shown in black, red, blue, and green lines, respectively), (**b**) the CDW and PHA yield of wild-type SG4502 and its recombinant strains.

**Figure 5 polymers-15-02290-f005:**
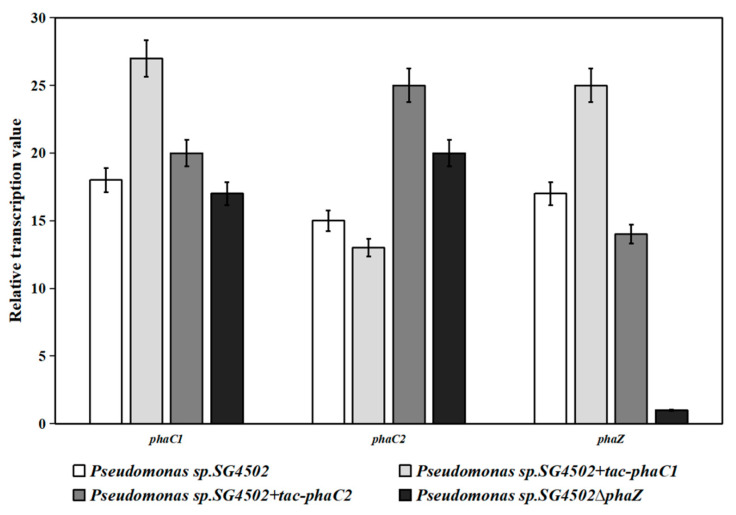
RT-qPCR analysis of *phaC1*, *phaC2* and *phaZ*. Error bars represent standard deviations (three replicates). Results of *Pseudomonas* sp. SG4502, *+*(*tac-phaC1*), *+*(*tac-phaC2*) and ∆*phaZ* are shown in white, light grey, dark grey and black, respectively.

**Figure 6 polymers-15-02290-f006:**
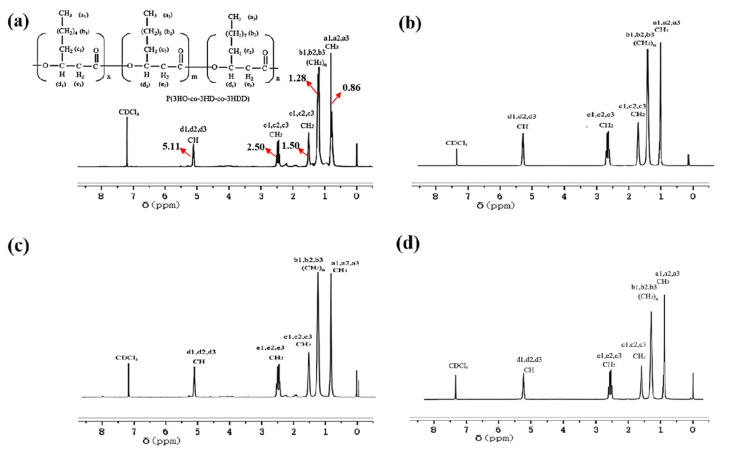
^1^H-NMR results of culture products of wild-type SG4502 (**a**), recombinant strain *+*(*tac-phaC1*) (**b**), *+*(*tac-phaC2*) (**c**) and ∆*phaZ* (**d**).

**Figure 7 polymers-15-02290-f007:**
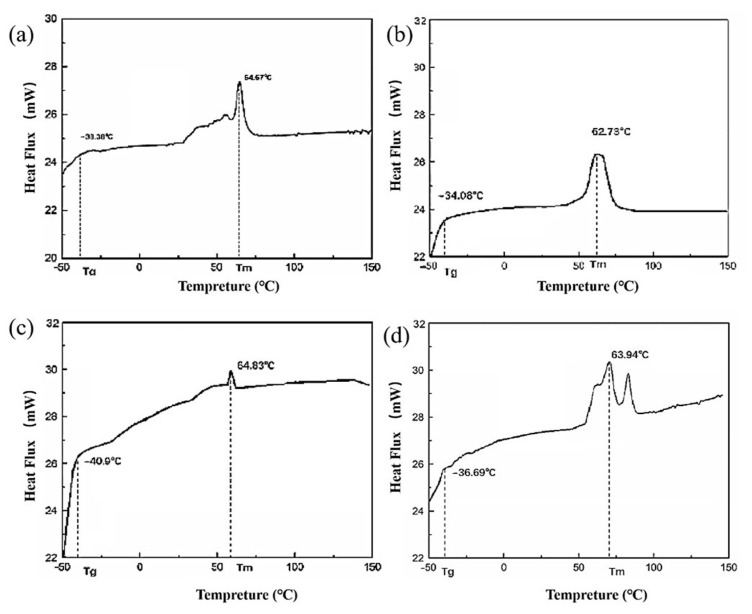
Differential scan heat result of (**a**) wild-type SG4502, (**b**) recombinant strain +(*tac-phaC1*), (**c**) recombinant strain +(*tac-phaC2*), (**d**) knockout strain ∆*phaZ*. *T_m_*: melting temperature, *T_g_*: glass transition temperature.

**Figure 8 polymers-15-02290-f008:**
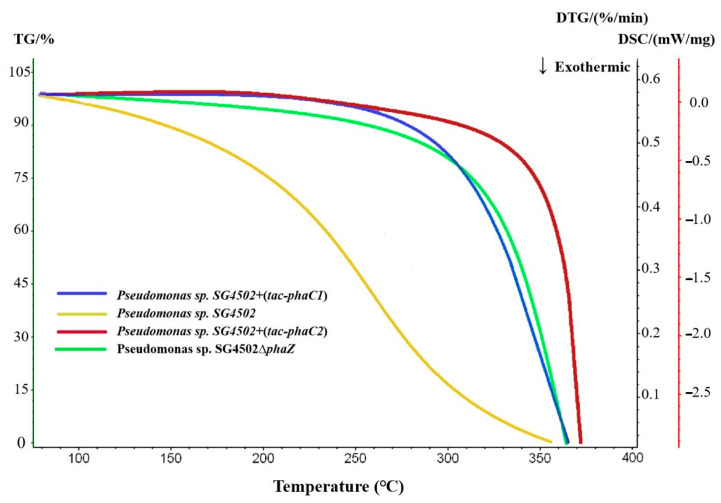
Thermogravimetric analysis results of culture products of wild-type SG4502, *+*(*tac-phaC1*), +(*tac-phaC2*) and ∆*phaZ*. (Results of wild-type SG4502, +(*tac-phaC1*), +(*tac-phaC2*) and ∆*phaZ* are shown in yellow, blue, red and green lines, respectively).

**Table 1 polymers-15-02290-t001:** GPC results of wild-type SG4502 and recombinant strains.

Strains	Number Average Molecular Weight (M_w_, Da)	Weight Average Molecular Weight (M_n_, Da)	Polydispersity Index (PDI) M_w_/M_n_
wild-type SG4502	3298	15,038	4.56
+(*tac-phaC1*)	4965	13,265	2.67
+(*tac-phaC2*)	5480	13,799	2.52
∆*phaZ*	7675	19,982	2.60

## Data Availability

Data is contained within the article or Appendix A.

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
