# Peer review of "Enhancing Production of Medium-Chain-Length Polyhydroxyalkanoates from Pseudomonas sp. SG4502 by tac Enhancer Insertion"

_polymers, 2023, doi:10.3390/polym15102290_

Round 1

Reviewer 1 Report

The study by Song and co-authors presents the genetic engineering of a Pseudomonas strain to produce different PHA. Biopolymers were produced and characterized using different methodologies. Please see comments below:

General comments

- Original studies or adequate reviews by experts should be cited instead of studies that focus on a different subject (e.g. references 1-3). Please consider citing the adequate literature.

- Discussion is simplistic, must be elaborated.

Specific comments

L 17-19. Not clear what are the two approaches. Please rewrite.

L 22-25. From which carbon source? Please clarify.

L 26. The most adequate terminology is size-exclusion chromatography - please correct according to the IUPAC https://goldbook.iupac.org/terms/view/S05705

L 25-32. Please consider adding the differences on the monomeric composition of the biopolymer produced by wild-type and recombinant strains.

L 41. A more adequate reference is doi: 10.1038/191463a0

L 44. A more adequate reference is 10.1042/BJ20031254

L 76-79. The production of scl-mcl was also reported when expressing Ralstonia eutropha or Aeromonas sp. phaC (doi.org/10.1159/000495752, doi.org/10.1016/j.ijbiomac.2018.03.051). Consider adding some studies in the introduction/ discussion section.

L 88. The expression of genes under the control of Ptac promoter - please rewrite.

L 91. Make sure all the scientific nomenclature is styled in italics (main text, figures, reference list etc).

L 96. 20% of cell dry weight (titer).

L 97. P. putida KT2442 is a mutant of P. putida KT2440, and not a wild type strain - 10.1073/pnas.78.12.7458

L 98. Please make sure it is yield (g/g), and not titer (g/L).

L 133. LB does not stand for Luria Bertani, please correct - doi.org/10.1128/JB.186.3.595-600.20

L 151. smr?

L 199. plasmids or PCR products obtained from genomic DNA?

L 227. Why was the mRNA isolated at 72 hours? Please present evidence indicating that the PHA accumulation period for the recombinant strains.

L 249. Please clarify what 'fermented' means.

L 302-303. Was it considered the construction of a mutant combining enhancer insertion and phaZ deletion?

L 316. Yield refers to grams of PHA produced from grams of carbon source. PHA concentration = titer (g/L). Please correct.

L 328. Figure 4. This is one of the most important results of the manuscript, please enhance figure quality.

L 337. Is there an explanation on why phaC1 level was also increased when the enhancer is located upstream of phaC2?

L 351-354. It is not immediately clear that the enhancer is located downstream of phaC2 in +(tac-phaC2) strain. I suggest adding a scheme with the three constructs indicating where the enhancer was introduced.

L 380-383. Can you estimate the mol% composition of each of the analyzed polymers? If not, I would suggest to analyze it using GC in the future.

Minor editing of English language required.

Reviewer 2 Report

Enhancing Production of Medium-chain-length Polyhydroxyalkanoates from Pseudomonas sp. SG4502 by tac Enhancer Insertion is well written except for few grammatical mistakes which authors may check again.

Few errors are pointed out below which may be corrected, and likewise other such errors may be removed:

Line 244, 250: deuterium chloroform reagent be replaced by Deuterated Chloroform

Line 253: Chromatographically pure chloroform be replaced by HPLC grade chloroform

Line 22, 23 and 24: 1H-NMR results showed that the products synthesized contained 3-hydroxyoctanoic acid (3HO), 3-hydroxydecanoic acid (3HD) and 3-hydroxydodecanoic acid (3HDD) units, which is consistent with those synthesized by the wild-type strain

Question:

PHAs being polyesters must be having carboxylic and hydroxyl groups at the terminal positions, why no acidic or hydroxyl group protons are not being shown in the 1H-NMR.

Round 2

Reviewer 1 Report

All the corrections/suggestions were incorporated, and manuscript quality is now adequate.

Minor comments

1. Please review the reference list. For example Kondo et al is not listed (should be number 16, Oliveira-Filho et al should be number 17.)

2. Standardize the used units - if using g/g use g/L; the other option is g.g-1 and g.L-1

 Minor editing of English language required
